# The Relationship of Cognitive Dysfunction with Inflammatory Markers and Carotid Intima Media Thickness in Schizophrenia

**DOI:** 10.3390/jpm13091342

**Published:** 2023-08-30

**Authors:** Okan İmre, Cuneyt Caglayan, Mehmet Muştu

**Affiliations:** 1Department of Psychiatry, Faculty of Medicine, Karamanoglu Mehmetbey University, Karaman 70200, Turkey; okanimre@kmu.edu.tr; 2Department of Medical Biochemistry, Faculty of Medicine, Bilecik Seyh Edebali University, Bilecik 11200, Turkey; 3Department of Cardiology, Faculty of Medicine, Karamanoglu Mehmetbey University, Karaman 70200, Turkey; drmustumehmet@kmu.edu.tr

**Keywords:** atherosclerosis, cognitive dysfunction, inflammation, schizophrenia

## Abstract

Objectives: Schizophrenia is a devastating and chronic mental disorder that affects 1% of the population worldwide. It is also associated with cognitive dysfunction and cardiovascular risk factors. The aim of this study is to investigate the relationship between cognitive impairment and some inflammatory markers and carotid intima-media thickness (CIMT) in schizophrenia. Methods: The participants of this study were 51 schizophrenia and 57 healthy controls (HC). The Positive and Negative Syndrome Scale (PANSS) was used for severity of illness, and the Montreal Cognitive Assessment Scale (MoCA) was used for cognitive functioning. The MoCA scores, some biochemical and inflammatory markers, and CIMT were compared between schizophrenia and HC groups. Results: Of the patients with schizophrenia, 11 were women (21.6%), and 40 were men (78.4%). MoCA scores were lower, and levels of NLR, MLR, PLR, SII, CRP, ESR, and CIMT were higher in schizophrenia compared to the HC group (respectively; *p* < 0.001, *p* < 0.001, *p* = 0.035, *p* = 0.008, *p* = 0.002, *p* < 0.001, *p* < 0.001, *p* < 0.001). In the schizophrenia group, there was no correlation between MoCA and inflammatory markers. MoCA and CIMT had a significant negative and moderate correlation (*p* < 0.001). Conclusions: This is the first study to show the relationship between cognitive impairment and CIMT in schizophrenia. In this study, NLR, MLR, PLR, SII, CRP, and ESR markers were higher in schizophrenia compared to HC, indicating inflammation. Our finding of elevated CIMT in schizophrenia suggests that there may be an atherosclerotic process along with the inflammatory process. The finding of a positive correlation between cognitive impairment and CIMT may be promising for new therapies targeting the atherosclerotic process in the treatment of cognitive impairment.

## 1. Introduction

Schizophrenia is a serious neuropsychiatric disorder that typically emerges in late adolescence, persists throughout life, and affects approximately 1% of the population [1]. It consists of three main symptom groups: positive, negative, and cognitive symptoms. The positive symptoms include delusions and hallucinations. The negative symptoms include alogia, anhedonia, avolition, and social withdrawal. Cognitive symptoms include problems in attention, verbal learning, processing speed, problem-solving, working memory, visuospatial learning, and cognitive flexibility. Cognitive symptoms are observed at every stage or even before schizophrenia emerges, negatively affecting life and reducing the quality of life [2]. These symptoms have been observed in 80% of schizophrenia patients [3] and are responsible for the majority of dysfunctions in schizophrenia [4]. While current treatments are effective for positive and negative symptoms, they are not proven to be effective for cognitive symptoms [5] owing to the limited information on the etiopathogenesis of cognitive impairment. Studies on the etiopathogenesis of cognitive impairment in schizophrenia have reported that cognitive impairment is associated with inflammation, and various biochemical markers of cognitive impairment have been suggested [6]. However, despite these studies, there is no definite cure that could be used as a treatment target [7].

Biomarkers can provide information about the etiopathogenesis of a disease and are important for determining therapeutic targets, follow-up, and prognosis. In previous studies, neutrophil/lymphocyte ratio (NLR), monocyte/lymphocyte ratio (MLR), platelet/lymphocyte ratio (PLR), and Systemic Immune-Inflammation Index (SII) have been suggested as inflammatory markers [8,9]. C-reactive protein (CRP) is among the most sensitive biomarkers of peripheral inflammation. It is produced by hepatocytes in the liver and is greatly increased during any type of inflammation [10]. Erythrocyte sedimentation rate (ESR) is a simple and inexpensive laboratory test used to assess inflammation or acute phase response [11].

In addition, it has been reported that carotid intima-media thickness (CIMT) represents the peak of ongoing subclinical inflammation and, therefore, may be an indicator of inflammation. Since the carotid intima-media thickness is measured non-invasively, it can be obtained from peripheral blood biomarkers more easily. Some studies have investigated the relationship between cognitive function and peripheral blood inflammatory markers in patients with schizophrenia [12,13,14]. However, no studies have investigated the relationship between cognitive impairment and CIMT in patients with schizophrenia.

In this study, we aimed to determine whether there was a difference in the cognitive functions, biochemical parameters, and carotid intima-media thicknesses of patients with schizophrenia and HC. In addition, we aimed to investigate the relationship between cognitive function scores in schizophrenic patients, biochemical parameters, and CIMT values and to determine whether there is a marker that may be related to cognitive impairment in patients with schizophrenia.

## 2. Material and Methods

### 2.1. Ethical Approval and Participants

This was a case–control study. The patient group consisted of schizophrenic patients between the ages of 18–65 who were followed up and treated at the Psychiatry Clinic of the Karaman Training and Research Hospital. According to the Diagnostic and Statistical Manual of Mental Disorders, Fifth Edition (DSM-5-TR), all patients met the criteria for schizophrenia [15]. Ethical approval was obtained for the study from the Karamanoglu Mehmetbey University Faculty of Medicine Clinical Research Ethics Committee (Approval number: 05-2023/02).

*Exclusion criteria:* All endocrinological diseases, hypertension, cardiovascular disease, inflammatory diseases, neurological diseases, metabolic syndrome, hyperlipidemia, hearing and speech problems, inability to cooperate with the test, lack of informed consent, and use of drugs other than psychiatric drugs were excluded from the study. Since substance addiction is a confounding factor for cognitive impairment, substance users in the patient and health control groups were not included in the study. Of the 80 patients with schizophrenia who were followed up regularly in our clinic, 29 were not included in the study as they did not meet the study criteria. Among the schizophrenic patients excluded from the study, three had heart failure, five had hypertension, one had hearing problems, three had kidney failure, and 27 had metabolic syndrome and various endocrinological problems. Two people were excluded from the study due to issues with cooperation during the test. A total of 51 studies with 51 patients with schizophrenia (11 women and 40 men) met the criteria.

This study included 57 HC participants (17 women and 40 men) matched for age, sex, and BMI. In the HC group, those with a history of any psychiatric illness and the exclusion criteria listed above were also excluded from the study. This study was performed in accordance with the guidelines of the Declaration of Helsinki. Written informed consent was obtained from all patients after the study procedure was explained. Informed consent was obtained from the guardians of patients with schizophrenia who were unable to provide written informed consent. All HCs agreed to participate in the study and provided written voluntary informed consent.

Sociodemographic data of patients with schizophrenia were recorded for the first time. The blood samples were collected after 8 h of fasting. ESR level and hemograms were measured in the EDTA tubes (Mindray, BC-6000, Shenzhen, China). The levels of NLR, MLR, PLR, and SII index were calculated from these data. CRP level was measured in the serum (Beckman Coulter, AU5800, Pasadena, CA, USA). The severity of symptoms in patients with schizophrenia was measured using the Positive and Negative Syndrome Scale (PANSS) by a psychiatrist. Cognitive function was also evaluated using the Montreal Cognitive Assessment Scale (MoCA). Carotid intima-media thickness was measured by a specialist cardiologist using an ultrasound device. Similar procedures were applied for the healthy control group as well.

### 2.2. Systemic Immune-Inflammation Index (SII)

SII Index, which is accepted as a new inflammation index, is a value calculated using lymphocyte, neutrophil, and platelet counts. SII = platelet count × neutrophil/lymphocyte count [8].

### 2.3. The Positive and Negative Syndrome Scale (PANSS)

The PANSS method was developed by Kay et al. [16]. This scale has three subcategories: positive (SAPS), negative (SANS), and general psychopathology. A trained specialist measured the severity of the disease by grading 30 different symptoms from 1 to 7 on this scale. This scale, designed as a clinical interview, takes approximately 40–50 min to complete. The total score obtained from the PANSS was a minimum of 30 and a maximum of 210. A high score indicates the high severity of the disease [16].

### 2.4. Montreal Cognitive Assessment Scale (MoCA)

The MoCA was developed by Dr. Ziad Nasreddine in Montreal, Canada, for the detection of mild cognitive impairment (MLB) [17]. Short-term memory, attention, working memory, and executive function were assessed with the MoCA test. The evaluation consists of a 30-point test that can be administered in 10–15 min. Cognitive function was considered normal in individuals with a score of 26 or above. A score of 25 or less on the MoCA indicates cognitive impairment. MoCA has been reported to be more sensitive than other cognitive tests for the assessment of cognitive function in patients with schizophrenia [18].

### 2.5. Carotid Intima Media Thickness (CIMT) Measurement

The High-resolution B-mode ultrasound images (Mindray DC-8 Exp Ultrasonography Device,1.1–4.4 MHz Single Crystal Phased Array Transducer, Shenzhen, China) with a 7.0 MHz linear array transducer (with 3.5 or 5 MHz transducers) were used to measure the CIMT. To measure CIMT, participants were placed supine, and their necks were kept in extension. Measurements were performed at a distance of 1 cm from the bifurcation of the external and internal carotid artery branches. CIMT was measured in the posterior, anterior, and lateral projections. The mean wall thickness of the common carotid artery was used as a key variable for the statistical analysis due to its strong relation with cardiovascular risk factors.

Cognitive functions, blood parameters, and CIMT of the schizophrenia and HC groups were compared within the scope of this study. Following that, the relationship between the cognitive function (MoCA scores) of patients with schizophrenia and sociodemographic data, inflammatory markers, and CIMT were statistically analyzed.

### 2.6. Statistical Analysis

All data were analyzed using the SPSS 25.0 package program. Variables were summarized as frequency “n”, percent “%”, arithmetic mean “$\bar{X}$”, and standard deviation “Sd.” Categorical data were compared using the chi-squared test. The conformity of continuous data to normal distribution was evaluated using the Kolmogorov–Smirnov test, q-q plot, skewness, and kurtosis. The research variables met the normality assumption. The independent group *t*-test was used to analyze independent paired groups. Pearson’s correlation test was used to evaluate the relationship between numerical measurements. Interpretation of the correlation coefficient was evaluated as (Weak = 0.01–0.49; Medium = 0.50–0.69; High = 0.70–1.00). For all analysis results, a significance level of *p* < 0.05 was accepted.

## 3. Results

Of the schizophrenic patients, 11 (21.6%) were women, 40 (78.4%) were men, and 66.7% of these patients were single. The mean illness duration was 22.7 years. The mean age was 47.16 ± 10.23 years. Of these, 43.1% were smokers, and 41.2% had a family history of psychiatric illness. Of these patients, 43.1% were using depot antipsychotics, 49% were using antidepressants, and 23.5% were using mood stabilizers. The most commonly used antipsychotic was olanzapine (38.9%), and the most commonly used depot antipsychotic was paliperidone (50%) (Table 1). Sex and smoking status of the schizophrenia and HC groups did not statistically differ between the groups (*x*^2^ = 0.995; *p* = 0.328, *x*^2^ = 0.012; *p* = 0.914) (Table 2).

When the schizophrenia group was compared with the HC group, the mean MoCA (t = −10.91; *p* < 0.001), education years mean (t = −9.68; *p* < 0.001), and lymphocyte mean were low (t = −2.99; *p* = 0.004) while NLR (t = 4.16; *p* < 0.001), MLR (t = 2.15; *p* = 0.035), (t = 2.74; *p* = 0.008), SII (t = 3.19; *p* = 0.002), CRP (t = 6.60; *p* < 0.001), ESR (t = 3.35; *p* < 0.001), and CIMT means were high (t = 8.08; *p* < 0.001) (Table 3).

In the schizophrenia group, MoCA had a significant negative and weak correlation with SAPS (*p* = 0.013, r = −0.349), a significant negative and weak correlation with SANS (*p* = 0.007, r = −0.377), and a significant negative and moderate correlation with CIMT (*p* < 0.001, r = −0.549). SANS score was positively correlated with duration of disease, BMI, CRP, and CIMT (respectively, *p* = 0.031; *p* = 0.031; *p* = −0.019; *p* = 0.039; *p* < 0.001). No correlation was found between SAPS, CIMT, and inflammatory parameters (Table 4).

When the MoCA value in the schizophrenia group was compared according to smoking, using depot antipsychotics, using antidepressants, and using mood stabilizers, it was seen that there was no difference between the groups (respectively, t = 0.96; *p* = 0.343, t = −0.28; *p* = 0,784, t = 0.50; *p* = 0.619, t = 0.12; *p* = 0.907) (Table 5).

## 4. Discussion

In the first plan, MoCA scores were compared with the HC group to determine whether there was cognitive impairment in the schizophrenia group. In accordance with the literature, the schizophrenia group had lower cognitive functions. In addition, cognitive functions were observed to decrease in parallel to the severity of the disease increased. Secondly, to determine the presence of inflammation, if any, in the schizophrenia group and whether there were differences in the CIMT value, the data was compared with the existing parameters of the HC group. Inflammatory parameters such as NLR, MLR, PLR, CRP, and ESR were found to be higher in the schizophrenia group. CIMT was higher in the schizophrenia group. This supports the association between the disease and inflammation. In addition, CIMT was found to be higher, suggesting that there may be an atherosclerotic process in schizophrenia patients. Finally, when we investigated the relationship between cognitive impairment, inflammatory parameters, and CIMT in schizophrenia patients, which was the main idea of our study, we found that cognitive impairment was only associated with CIMT.

In our study, no relationship was found between the severity of positive symptoms in schizophrenia and inflammatory parameters or CIMT. However, a positive correlation was found between the severity of negative symptoms and CRP and CIMT. This may be because negative symptoms are closely related to cognitive impairment. As far as we know, there is no study in the literature on the relationship between the severity of negative symptoms and CIMT.

In our study, NLR, MLR, PLR, and SII inflammatory parameters were high in schizophrenia. There are many studies investigating the relationship between schizophrenia and inflammation parameters [19,20,21]. In the literature, there are consistent results in studies on NLR and SII. The results are inconsistent in terms of MLR and PLR. In a meta-analysis, NLR, MLR, and PLR levels were found to be higher in schizophrenia, similar to our study [9]. In a recent study, it was reported that NLR, MLR, and SII levels were higher in schizophrenia patients, but there was no difference in terms of PLR [22]. In another study, NLR, PLR, and SII were higher in the schizophrenia group, while no significant difference was found in terms of MLR [23].

In the current study, CRP was higher in the schizophrenia group as compared to the HC group. This finding is consistent with the literature. The effect of CRP on schizophrenia has been well-researched [24]. In a study, it was reported that CRP levels, which were higher in schizophrenia patients, decreased after treatment, and CRP was recommended as a prognostic marker [25]. In our study, ESR was higher in the schizophrenia group. There are few studies investigating the ESR level in schizophrenia. The majority of the studies were carried out decades ago. In a recent longitudinal study, it was reported that those with high ESR values had a higher risk of developing schizophrenia later [26]. Our study may contribute to the literature in this respect.

When cognitive impairment and inflammatory parameters were investigated in schizophrenia patients, which is the main idea of our study, cognitive impairment was not associated with the inflammatory parameters we examined. There are many studies in the literature investigating the relationship between cognitive impairment and inflammatory markers in schizophrenia [27,28,29,30]. However, when we searched for studies on cognitive impairment and our current parameters in schizophrenia, we could only find studies on NLR and CRP. We did not find any studies on MLR, PLR, SII, or ESR. In the limited number of studies on NLR, no relationship was found between NLR and cognitive functions in untreated psychosis patients [31]. There are many studies investigating the relationship between cognitive impairment and CRP in schizophrenia patients. In the majority of studies, elevated CRP was found to indicate poor cognitive performance [27,32].

Inflammation may play a role in the pathophysiology of schizophrenia by affecting several pathways, such as neurodegenerative processes, oxidative stress, immune response, and nerve conduction [33]. A myriad of studies have been undertaken to investigate the role of inflammation in the onset and progression of schizophrenia. In a meta-analysis of 41 studies, it was reported that IL-6 and CRP levels were consistently increased in all stages of schizophrenia, and elevated CRP was associated with more cognitive impairment [34]. Recent studies are promising for targeting specific disease subgroups in anti-inflammatory therapy. In a meta-analysis, various anti-inflammatory drugs were reported to be effective on positive, negative, and cognitive symptoms of schizophrenia. In this meta-analysis, adding minocycline and pregnenolone to treatment was found to be beneficial to cognitive functions [35]. However, further studies and large-scale meta-analyses to find the specific patient subgroup for personalized treatment may help us better understand this relationship.

In our study, the CIMT value, which may be related to inflammation, was higher in the schizophrenia group compared to the HC group. In the literature, we could find only two studies evaluating CIMT levels in schizophrenia patients. In both studies, higher CIMT values were reported in the schizophrenia group [36,37]. This supports the relationship between the disease and inflammation. However, increased CIMT has been reported to be a strong predictor of future cardiovascular events [38]. People with schizophrenia die 15 years earlier than the normal population. One of the main causes of these deaths is cardiovascular disease [39]. Early diagnosis of asymptomatic cardiovascular diseases in schizophrenia patients will improve their quality of life and reduce mortality.

In meta-analyses, it has been reported that patients with schizophrenia are at high risk for coronary heart disease, cerebrovascular disease, and congestive heart failure for many reasons, such as side effects of the drugs used, metabolic syndrome, inactivity, and smoking [40].

Therefore, it is important to know the relationship between schizophrenia and cardiovascular disease. In some studies, it has been stated that the risk of cardiovascular disease is still high, although conditions that increase the risk of cardiovascular disease, such as medications, diet, and inactivity, are excluded. This may be due to a common genetic structure. In a study, it was reported that people with a high polygenic risk score for schizophrenia had a decrease in heart volume, an increase in myocardial stiffness, and an increase in ejection fractions [41]. Studies have suggested high CIMT as a marker of subclinical atherosclerosis [42,43]. In schizophrenia patients with high CMT levels, a cardiology examination at regular intervals may be important for the early detection and treatment of cardiovascular diseases.

Recent studies have reported that atherosclerosis in the carotid artery may be an independent risk factor for cognitive dysfunction [44]. The mechanisms by which atherosclerosis impairs cognitive functions are not yet fully known [45], but some assumptions are put forward. The increased arterial pulsatile load may have affected neurocognitive functions by causing microvascular damage in the brain [46]. By slowing cerebral blood flow, carotid atherosclerosis may reduce oxygenation of the brain and lead to dementia-related atrophy [47]. Thickening of the carotid artery may lead to neurodegeneration in the hippocampus [48]. In addition, neurocognitive impairment may have developed as a result of hypoperfusion in the brain with silent embolism [49].

The high CIMT value found to be associated with cognitive impairment in schizophrenia in our study suggests that cognitive impairment may also be associated with atherosclerotic processes; if subclinical atherosclerosis can be predicted with CIMT, this may provide an earlier intervention opportunity to slow cognitive dysfunction [50]. This finding could identify both groups at risk of cardiovascular disease and those with cognitive impairment and guide new treatment targets in these patient subgroups. If our findings are supported by new studies, CIMT can be used as a potential marker for both cognitive impairment and cardiovascular disease risk in patients with schizophrenia. Further studies and large-scale meta-analyses may help us better understand this relationship.

## 5. Limitations of the Study

The study has some limitations. We had a small number of patients. Other parameters that may be related to inflammation were not studied. More information can be obtained by using a more comprehensive inflammation panel, examining cytokines, oxidative stress, and similar biomarkers, and conducting neuroimaging studies.

## 6. Conclusions

In this study, higher levels of NLR, MLR, PLR, SII, CRP, and ESR markers in schizophrenia patients compared to the control group indicate inflammation. Our finding of elevated CIMT in schizophrenia patients suggests that there may be an atherosclerotic process along with the inflammatory process. The finding of a positive correlation between cognitive impairment and CIMT may be promising for new therapies targeting the atherosclerotic process in the treatment of cognitive impairment. Large-scale studies are needed for this.

## Figures and Tables

**Table 1 jpm-13-01342-t001:** Sociodemographic data of the schizophrenia group.

		n	%
Gender	Female	11	21.6
	Male	40	78.4
Marriage Status	Single	34	66.7
	Married	5	9.8
	Divorced	12	23.5
Smoker	Yes	22	43.1
	No	29	56.9
Psychiatric disorder in the family	Yes	21	41.2
	No	30	58.8
Depot antipsychotic	Yes	22	43.1
	No	29	56.9
Depot antipsychotic type	Paliperidone	11	50.0
	Zuclopenthixol	6	27.3
	Risperidone	4	18.2
	Aripiprazole	1	4.5
Antidepressant	Yes	25	49.0
	No	26	51.0
Mood stabilizer	Yes	12	23.5
	No	39	76.5
Mood stabilizer type	Valproic acid	8	61.5
	Lithium	3	23.1
	Lamotrigine	1	7.7
	Carbamazepine	1	7.7
Antipsychotic type	Olanzapine	28	38.9
	Quetiapine	12	16.7
	Clozapine	10	13.9
	Aripiprazole	7	9.7
	Amisulpride	7	9.7
	Risperidone	4	5.6
	Haloperidol	2	2.8
	Paliperidone	1	1.4
	Zuclopenthixol	1	1.4

**Table 2 jpm-13-01342-t002:** Comparison of groups by gender and smoking status.

			Schizophrenia	HC	Total	*x* ^2^	* *p*
Gender	Female	n (%)	11 (21.6)	17 (29.8)	28 (25.9)	0.955	0.328
	Male	n (%)	40 (78.4)	40 (70.2)	80 (74.1)		
Smoker	Yes	n (%)	22 (43.1)	24 (42.1)	46 (42.6)	0.012	0.914
	No	n (%)	29 (56.9)	33 (57.9)	62 (57.4)		

* Chi-square Test (HC: Healthy control).

**Table 3 jpm-13-01342-t003:** Comparison of various parameters of schizophrenia and HC group.

								95% CI of the Difference		
		n	$\bar{X}$	Sd	t	Md		Lowest Limit	Md	* *p*
MoCA	SCH	51	18.06	6.41	−10.91	−9.91		−11.73	−8.08	<0.001
	HC	57	27.96	1.07						
Age	SCH	51	47.16	10.23	0.47	0.84		−2.72	4.40	0.640
	HC	57	46.31	8.14						
Education years	SCH	51	8.24	4.24	−9.68	−6.33		−7.63	−5.02	<0.001
	HC	57	14.56	2.06						
BMI	SCH	51	28.18	6.95	1.56	1.78		−0.48	4.04	0.122
	HC	57	26.41	4.29						
WBC	SCH	51	7.38	2.03	−0.31	−0.11		−0.81	0.59	0.756
	HC	57	7.49	1.62						
Neutrophil	SCH	51	4.81	1.66	1.45	0.40		−0.15	0.94	0.152
	HC	57	4.41	1.11						
Lymphocyte	SCH	51	1.94	0.78	−2.99	−0.42		−0.69	−0.14	0.004
	HC	57	2.35	0.67						
Monocyte	SCH	51	0.45	0.17	−1.43	−0.04		−0.10	0.02	0.156
	HC	57	0.49	0.14						
Platelet	SCH	51	233.35	60.25	−1.39	−13.95		−33.89	5.99	0.168
	HC	57	247.30	40.98						
NLR	SCH	51	2.86	1.40	4.16	0.88		0.46	1.31	<0.001
	HC	57	1.98	0.60						
MLR	SCH	51	0.26	0.13	2.15	0.04		0.00	0.08	0.035
	HC	57	0.22	0.06						
PLR	SCH	51	135.78	52.16	2.74	23.29		6.38	40.20	0.008
	HC	57	112.49	32.87						
SII	SCH	51	653.97	348.80	3.19	168.94		63.31	274.56	0.002
	HC	57	485.03	153.78						
CRP	SCH	51	6.43	5.32	6.60	5.05		3.51	6.59	<0.001
	HC	57	1.38	1.06						
ESR	SCH	51	9.94	10.52	3.35	5.18		2.08	8.29	<0.001
	HC	57	4.76	3.22						
CIMT	SCH	51	0.98	0.19	8.08	0.23		0.18	0.29	<0.001
	HC	57	0.75	0.09278						

* *t*-Test in Independent Groups, Md: The mean difference, Sd: Standard deviation, MoCA: Montreal cognitive assessment scale, BMI: Body mass index, NLR: neutrophil/lymphocyte ratio, PLR: platelet/lymphocyte ratio, MLR: monocyte/lymphocyte ratio, SII: Systemic Immune-Inflammation Index, CRP: C-reactive protein, ESR: erythrocyte sedimentation rate, CIMT: Carotid Intima-Media Thickness.

**Table 4 jpm-13-01342-t004:** Comparison of moCA and PANSS scores with other parameters in schizophrenia.

		SAPS	SANS	Age	İllnessduration	BMİ	WBC	Neutrophil	Lymphocyte	Monocyte	Platelet	CRP	ESR	NLR	MLR	PLR	SII	CIMT
MoCA	r	−0.349	−0.377	−0.391	−0.274	0.154	0.171	0.092	0.259	−0.033	0.018	0.009	−0.121	−0.122	−0.232	−0.265	−0.107	−0.549
	*p*	0.013	0.007	0.005	0.052	<0.001	0.286	0.230	0.521	0.066	0.816	0.951	0.401	0.395	0.102	0.060	0.454	<0.001
SAPS	r		−0.383	0.040	−0.187	−0.205	−0.166	−0.118	−0.147	−0.081	0.014	0.061	0.040	0.128	0.168	0.201	0.089	0.017
	*p*		0.006	0.783	0.193	0.158	0.250	0.413	0.307	0.576	0.923	0.680	0.787	0.377	0.245	0.161	0.539	0.906
SANS	r			0.103	0.305	−0.309	−0.001	0.092	−0.240	0.151	−0.128	−0.296	−0.036	0.208	0.248	0.222	0.155	0.467
	*p*			0.476	0.031	0.031	0.993	0.526	0.093	0.295	0.374	0.039	0.808	0.148	0.082	0.121	0.281	<0.001

SAPS: Scale for the Assessment of Positive Symptoms, SANS: Scale for the Assessment of Negative Symptoms, PANSS: Positive and Negative Syndrome Scale, moCA: Montreal Cognitive Assessment scale, BMI: Body mass index, NLR: neutrophil/lymphocyte ratio, PLR: platelet/lymphocyte ratio, MLR: monocyte/lymphocyte ratio, SII: Systemic Immune-Inflammation Index, CRP: C-reactive protein, ESR: erythrocyte sedimentation rate, CIMT: Carotid Intima-Media Thickness.

**Table 5 jpm-13-01342-t005:** Comparison of MoCA in the schizophrenia group with other conditions.

							95% CI of the Difference		
		n	$\bar{X}$	Sd	t	Md	Lowest Limit	Highest Limit	* *p*
Smoker	Yes	22	19.05	5.51	0.96	1.74	−1.91	5.38	0.343
	No	29	17.31	7.02					
Depot_antipsychotic	Yes	22	17.77	6.14	−0.28	−0.50	−4.18	3.17	0.784
	No	29	18.28	6.70					
Antidepressant	Yes	25	18.52	6.52	0.50	0.90	−2.73	4.54	0.619
	No	26	17.62	6.39					
Mood_stabilizer	Yes	12	18.25	6.73	0.12	0.25	−4.04	4.54	0.907
	No	39	18.00	6.39					

* *t*-Test in Independent Groups.

## Data Availability

Not applicable.

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
