# Peer review of "The Relationship of Cognitive Dysfunction with Inflammatory Markers and Carotid Intima Media Thickness in Schizophrenia"

_jpm, 2023, doi:10.3390/jpm13091342_

Round 1

Reviewer 1 Report

The authors' study looks at the relationship between cognitive impairment and 12 inflammatory markers, as well as carotid intima-media thickness (CIMT), in people with schizophrenia. The study includes 51 patients with schizophrenia and 57 healthy controls (HC). The Positive and Negative Syndrome Scale (PANSS) was used to assess the severity of the illness, while the Montreal Cognitive Assessment Scale (MoCA) was used to assess cognitive performance. The schizophrenia and HC groups were compared in terms of MoCA scores, biochemical and inflammatory markers, and CIMT.

 The findings highlight the significance of future research into the potential linkages between cognitive functioning and cardiovascular health in this population.

The manuscript presents several significant drawbacks that require attention from the authors before it can be considered for publication.

The paper is well-organized, and the findings are well-supported by statistical analysis. However, it would be advantageous if the authors could investigate further the underlying mechanisms that lead to the observed correlations between MoCA scores and CIMT.

What criteria did the authors utilize to classify or detect the severity of schizophrenia in the patients? Additionally, there is a consistent use of commas instead of decimal points in all tables. Kindly review and replace them accordingly.

The Figure 1 graph is absolutely confusing. There is no proper explanation given by authors in legends as well as in the main text. Please fix this issue. 

Have authors considered substance abuse as a confounding factor for cognitive decline? 

Did the authors examine the genetic risk of developing schizophrenia and cardiovascular disease in the participants?

Could you please explain how the Montreal Cognitive Assessment Scale (MoCA) was used to assess cognitive functioning? Were there any difficulties in utilizing this measure with persons who had schizophrenia?

In the study, how were inflammatory markers and CIMT tested or assessed? Were there any difficulties with the measurement's dependability or accuracy?

Were there significant changes in the distribution of CIMT and inflammatory markers between schizophrenia subgroups (e.g., symptom severity, treatment history)?

Can you go into further detail about the potential clinical significance of the findings on the relationship between cognitive impairment and CIMT in schizophrenia?

Were any changes performed in the statistical analysis for multiple comparisons to allow for potential false positives? 

Furthermore, a more in-depth discussion of the implications of these findings for clinical management and prospective therapeutic approaches for schizophrenia patients could significantly improve the study's value. 

Please check the grammar in the abstract. 

Author Response

Manuscript Number: jpm-2548178R1

Manuscript Title       : The Relationship of Cognitive Dysfunction with Inflammatory Markers and Carotid Intima Media Thickness in Schizophrenia

Thanks very much for taking your time to review this manuscript. I really appreciate all your comments and suggestions! Please find my itemized responses in below and my revisions/corrections in the re-submitted files. The revised parts of the article were painted in yellow.

We would like also to thank you for allowing us to resubmit a revised copy of the manuscript.

We hope that the revised manuscript is accepted for publication in the Journal of Personalized Medicine.

Responses to Reviewer-1

The authors' study looks at the relationship between cognitive impairment and 12 inflammatory markers, as well as carotid intima-media thickness (CIMT), in people with schizophrenia. The study includes 51 patients with schizophrenia and 57 healthy controls (HC). The Positive and Negative Syndrome Scale (PANSS) was used to assess the severity of the illness, while the Montreal Cognitive Assessment Scale (MoCA) was used to assess cognitive performance. The schizophrenia and HC groups were compared in terms of MoCA scores, biochemical and inflammatory markers, and CIMT.

 The findings highlight the significance of future research into the potential linkages between cognitive functioning and cardiovascular health in this population.

The manuscript presents several significant drawbacks that require attention from the authors before it can be considered for publication.

The paper is well-organized, and the findings are well-supported by statistical analysis. However, it would be advantageous if the authors could investigate further the underlying mechanisms that lead to the observed correlations between MoCA scores and CIMT.

RESPONSE: Many thanks to the reviewer because of his/her positive opinion. In line with your suggestions, information on the underlying mechanisms leading to the observed correlations between MoCA scores and CIMT has been added to the discussion section.

What criteria did the authors utilize to classify or detect the severity of schizophrenia in the patients? Additionally, there is a consistent use of commas instead of decimal points in all tables. Kindly review and replace them accordingly.

RESPONSE: The severity of symptoms in patients with schizophrenia was measured using the Positive and Negative Syndrome Scale (PANSS) by a psychiatrist. This scale has three subcategories: positive(SAPS), negative(SANS), and general psychopathology. A trained specialist measured the severity of the disease by grading 30 different symptoms from 1 to 7 on this scale. We used commas instead of decimal places in tables. We fixed these.

The Figure 1 graph is absolutely confusing. There is no proper explanation given by authors in legends as well as in the main text. Please fix this issue. 

RESPONSE: The Figure 1 has been omitted from the article to avoid confusion. It was included in the article for additional information only. Removing it will not cause any trouble.

Have authors considered substance abuse as a confounding factor for cognitive decline? 

RESPONSE: Thank you very much for reporting our mistake. We forgot to add it to the exclusion criteria in the method section. "Substance users in the patient and health control groups were not included in the study, as substance abuse is a confounding factor for cognitive impairment." We included this in the manuscript.

Did the authors examine the genetic risk of developing schizophrenia and cardiovascular disease in the participants?

RESPONSE: Dear Reviewer, we have only added literature information to the discussion section about this section. "People with schizophrenia die 15 years earlier than the normal population. One of the main causes of these deaths is cardiovascular disease (Laursen 2014). Therefore, it is important to know the relationship between schizophrenia and cardiovascular disease. In some studies, it has been stated that the risk of cardiovascular disease is still high, although conditions that increase the risk of cardiovascular disease such as medications, diet, and inactivity are excluded. This may be due to a common genetic structure. In a study, it was reported that people with a high polygenic risk score for schizophrenia had a decrease in heart volume, an increase in myocardial stiffness, and an increase in ejection fractions (Pillinger, 2023)."

Could you please explain how the Montreal Cognitive Assessment Scale (MoCA) was used to assess cognitive functioning? Were there any difficulties in utilizing this measure with persons who had schizophrenia?

RESPONSE: MoCA was administered by a specialist psychiatrist to both schizophrenia and healthy control groups. Severe patients who could not cooperate with the tests and those with active psychotic processes were excluded from the study, so no difficulties were experienced.

In the study, how were inflammatory markers and CIMT tested or assessed? Were there any difficulties with the measurement's dependability or accuracy?

RESPONSE: Thank you so much. After 8 hours of fasting, the blood of the participants was drawn. Blood samples were placed in EDTA tubes. Neutrophil, monocyte, platelet and lymphocyte count values were measured in an automatic whole blood analyzer. NLR, MLR, PLR and SII were calculated. CRP and ESR were checked.

Calculation of CIMT

First, the MoCA scores of the participants were calculated by a specialist psychiatrist. Then the participants were directed to the cardiology department. CIMT was measured from three sites procedurally by a specialist cardiologist who had no knowledge of the participants' MoCA score. MoCA and mean CIMT were then compared by a third party. No difficulties were encountered during the CIMT measurement.

Were there significant changes in the distribution of CIMT and inflammatory markers between schizophrenia subgroups (e.g., symptom severity, treatment history)?

RESPONSE: In our study, no relationship was found between the severity of positive symptoms in schizophrenia and inflammatory parameters or CIMT. However, a positive correlation was found between the severity of negative symptoms and CRP and CIMT. As far as we know, there is no study in the literature on the relationship between the severity of negative symptoms and CIMT.

Can you go into further detail about the potential clinical significance of the findings on the relationship between cognitive impairment and CIMT in schizophrenia?

 RESPONSE: Dear Reviewer, we have added information to the discussion section about this section. “In conclusion, if subclinical atherosclerosis can be predicted with CIMT, this may provide an earlier intervention opportunity to slow cognitive dysfunction (Lin,2020). At the same time, CIMT can predict the identification of schizophrenic patients at risk for cardiovascular disease. If our findings are supported by new studies, CIMT can be used as a potential marker for both cognitive impairment and cardiovascular disease risk in patients with schizophrenia.”

Were any changes performed in the statistical analysis for multiple comparisons to allow for potential false positives? 

RESPONSE: As the p-value was much lower than 0.05 in the analysis, it was thought that there was no risk of false positives.

Furthermore, a more in-depth discussion of the implications of these findings for clinical management and prospective therapeutic approaches for schizophrenia patients could significantly improve the study's value. 

RESPONSE: Thank you very much for your positive contributions. New information on this topic has been added to the discussion section.

Please check the grammar in the abstract.

RESPONSE: In general, the language of the article was checked by an expert whose native language is English.

Reviewer 2 Report

The article contains original and interesting data. The authors attempt to link inflammation biomarkers with parameters associated with cognitive deficits in patients with schizophrenia and healthy individuals.

I miss more advanced, multidimensional statistical methods, e.g. PCA. Also, the abbreviation of the MoCA scale should be used consistently in the same way.

Author Response

Manuscript Number: jpm-2548178R1

Manuscript Title       : The Relationship of Cognitive Dysfunction with Inflammatory Markers and Carotid Intima Media Thickness in Schizophrenia

Thanks very much for taking your time to review this manuscript. I really appreciate all your comments and suggestions! Please find my itemized responses in below and my revisions/corrections in the re-submitted files. The revised parts of the article were painted in yellow.

We would like also to thank you for allowing us to resubmit a revised copy of the manuscript.

We hope that the revised manuscript is accepted for publication in the Journal of Personalized Medicine.

Responses to Reviewer-2

The article contains original and interesting data. The authors attempt to link inflammation biomarkers with parameters associated with cognitive deficits in patients with schizophrenia and healthy individuals.

RESPONSE: Many thanks to the reviewer because of his/her positive opinion.

I miss more advanced, multidimensional statistical methods, e.g. PCA. Also, the abbreviation of the MoCA scale should be used consistently in the same way.

RESPONSE: Dear Reviewer, thank you very much for your positive suggestions. In line with your suggestions, we again met with an expert biostatistician. He decided that statistical analysis was appropriate as such. Therefore, we did not change the statistics data. Best regards.

“All data were analyzed using the SPSS 25.0 package program. Variables were summarized as frequency “n,” percent “%”, arithmetic mean “X̄”, standard deviation “Sd.” Categorical data were compared using the chi-squared test. Conformity of con-tinuous data to normal distribution was evaluated using the Kolmogorov-Smirnov test, q-q plot, skewness, and kurtosis. The research variables met the normality as-sumption. The independent group t-test was used to analyze independent paired groups. Pearson’s correlation test was used to evaluate the relationship between nu-merical measurements. Interpretation of the correlation coefficient was evaluated as (Weak =0.01-0.49; Medium =0.50-0.69; High =0.70-1.00). For all analysis results, a sig-nificance level of p <0.05 was accepted.”

Round 2

Reviewer 1 Report

The authors have provided appropriate responses to my questions and also made necessary changes. I endorse the acceptance of this paper.